# Automatic Control and Model Verification for a Small Aileron-Less Hand-Launched Solar-Powered Unmanned Aerial Vehicle

**An Guo [1], Zhou Zhou [1,*], Xiaoping Zhu [2], Xin Zhao [1] and Yuxin Ding [1]**

[1] School of Aeronautics, Northwestern Polytechnical University, Xi'an 710072, China; guoanuav@mail.nwpu.edu.cn (A.G.); xin_zhao@mail.nwpu.edu.cn (X.Z.); dingyuxin@mail.nwpu.edu.cn (Y.D.)

[2] UAV Research Institute, Northwestern Polytechnical University, Xi'an 710065, China; zhuxp@nwpu.edu.cn

* Correspondence: zhouzhou@nwpu.edu.cn

**Abstract:** This paper describes a low-cost flight control system of a small aileron-less hand-launched solar-powered unmanned aerial vehicle (UAV). In order to improve the accuracy of the whole system model and quantify the influence of each subsystem, detailed modeling of UAV energy and a control system including a solar model, engine, energy storage, sensors, state estimation, control law, and actuator module are established in accordance with the experiment and component principles. A whole system numerical simulation combined with the 6 degree-of-freedom (DOF) simulation model is constructed based on the typical mission route, and the parameter precision sequence and energy balance are obtained. Then, a hardware-in-the-loop (HIL) experiment scheme based on the Stewart platform (SP) is proposed, and three modes of acceleration, angular velocity, and attitude are designed to verify the control system through the inner and boundary states of the flight envelope. The whole system scheme is verified by flight tests at different altitudes, and the aerodynamic force coefficient and sensor error are corrected by flight data. With the increase of altitude, the cruise power increases from 47 W to 78 W, the trajectory tracking precision increases from 23 m to 44 m, the sensor measurement noise increases, and the bias decreases.

**Keywords:** solar-powered UAV; flight control system; Stewart platform; hardware-in-the-loop simulation; model calibration

## 1. Introduction

In recent decades, solar-powered unmanned aerial vehicles (UAVs) have attracted the attention of many research groups all over the world [1]. This new type of UAV consumes electrical energy converted from solar irradiance by solar cells mounted on the main wing. If the photovoltaic (PV) cells and battery produce and store sufficient extra energy during the daytime for nighttime flight, the UAVs could possibly fly for a virtually unlimited duration [2]. With the perpetual flight capability, they are great candidates for remote data collection or distribution [3]. Besides, large-scale disaster relief support missions, forest fire prevention, communication overlay, and endangered animal protection would benefit in particular from this long time and low-cost flight capability [4].

Unlike the large-scale high-altitude long-endurance (HALE) UAVs, smaller-scale solar-powered UAVs are mostly designed for the low-altitude long-endurance (LALE) flight, and they are not limited by take-off and landing conditions; thus, they have great application potential. However, these UAVs have to deal with more challenges including complex meteorological conditions, flight area restrictions, and communication interferences, but they provide the advantages of higher resolution imaging, lower hardware requirements, and lower costs. SoLong was the first-ever hand-launched solar-powered

UAV, which performed a continuous 48-hour flight but required eight pilots to perform the flight [5]. SkySailor UAV was proposed by Noth [6], which could fly autonomously except during take-off and landing, and the pilots were still needed. AtlantikSolar was a mature platform with a flight controller, mission payload, thermal updrafts autonomous tracking module, and it presented 81 h continuous flight [7]. Nowadays, challenges for LALE perpetual flight lies in transferring the technology from the concept or design stage into practice, and there are still plenty of details to be settled.

The energy-centered design criteria, aircraft shapes, solar cells, and light structure-integrated design, battery's energy density, as well as energy management and optimal trajectory control are key technologies for traditional large-scale solar-powered UAVs [8]. Motivated by Oettershagen's work, a complete control system development process is proposed for a 3 m wingspan and 3.1 kg aileron-less hand-launched solar-powered UAV (Figure 1), and the specific design requirements are: position accuracy within one square kilometer of 40 m/km$^2$, height accuracy of 10 m, maximum wind resistance of 8 m/s, the minimum temperature of –20 °C, and take-off and landing altitudes above 5000 m.

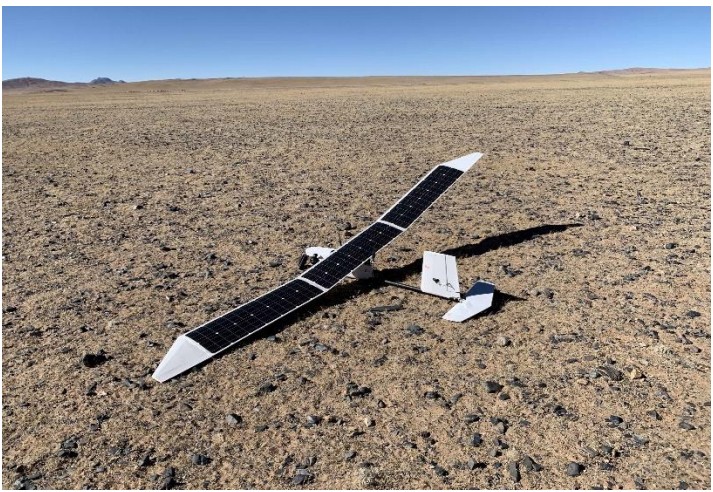

**Figure 1.** The low-altitude long-endurance (LALE) hand-launched solar-powered unmanned aerial vehicle (UAV) on the plateau.

The cost of a UAV platform and its operation is a crucial constraint to its application. At present, low-cost flight control platforms have been widely used in miniature air vehicles [9]. Smaller scale, lower-cost solar-powered hand-launched UAVs have the advantages of easier civilian application and marketization, yet there still might be a large amount of sensor errors, vulnerability to lower atmosphere influence, and flight instability [10]. Besides, the aileron-less design places a higher demand on the flight control system, and it is necessary to overcome the problems of only the rudder for lateral control and the lower measurement precision of a low-cost flight controller. Since the rudder not only performs the coordinated turning control, but also plays the role of the aileron, this leads to the roll and yaw control being significantly coupled, and the control law thus needs to be redesigned. Then, this indirect control scheme reduces the accuracy of trajectory tracking, and the measurement accuracy of the low-cost flight control is limited. Therefore, a filtering algorithm based on the characteristics of the UAV needs to be designed to improve the mission accuracy, detailed numerical and hardware-in-the-loop simulations (HILS) are also necessary for model error correction during the design process, and the accuracy of the model can be further improved by using flight data. This paper consequently aims to establish the state-of-the-art in the control system design of a hand-launched solar-powered UAV by introducing and extending relevant aspects from three levels, UAV systematic modeling, Stewart platform hardware-in-the-loop simulation, and lower and higher altitude flight tests, which together enable the whole scheme validation. The method from model establishment to verification is applied to the design of the control system for a real UAV, which shortens the development cycle, reduces the cost, and enables it to be applied in similar platforms.

This paper is organized as follows: Section 2 gives an overview of a whole component-level model including the energy and controller systems of the UAV; Section 3 describes the numerical simulation and the Stewart platform hardware-in-the-loop simulation verification schemes; Section 4 shows the field flight test results at different altitudes and at a model correction method; and finally, Section 5 gives a conclusion.

## 2. Complete System Modeling

The UAV adopts a normal configuration, without landing gear and aileron, and the wing adopts a lightweight structural design with the PV cells cover the upper skin. Besides, the imaging equipment is mounted on the nose in order to get a wide field view as well as avoid touchdown collision, and the engine system is placed on the rear side of the wing to generate thrust, as shown in Figure 2.

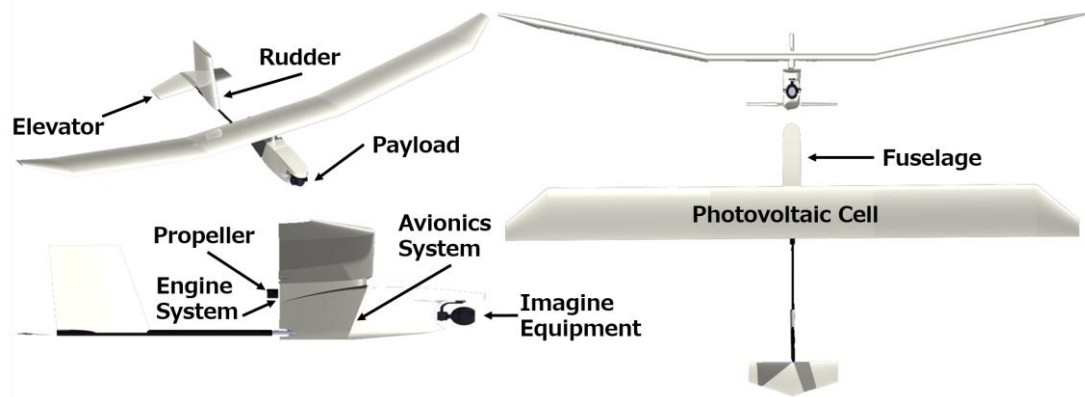

**Figure 2.** The airframe of a solar-powered UAV.

This lightweight and maximum energy harvesting design requires the UAV to keep a low speed ensuring a long endurance, and the aileron-less design reduces the difficulty of solar cell installation and improves the efficiency of energy production, but it also leads to a different control strategy compared with the normal configuration. In addition, the basic flight modes of the UAV are divided into hand-launched take-off, autonomous cruise, slower glide, and fuselage touching the ground. In the simulation, a 6 degree-of-freedom (DOF) equation of motion derived from [11] is applied for the UAV airframe modeling, and the overall parameters of the airplane are shown in Table 1.

**Table 1.** Summary of UAV design and performance characteristics.

| Parameters | Value | Unit |
|---|---|---|
| Wingspan | 3.2 | m |
| Chord length | 0.3 | m |
| Maximum weight | 3.6 | kg |
| Payload | 0.5 | kg |
| Battery weight | 0.6 | kg |
| Battery energy density | 270 | Wh/kg |
| Maximum speed | 32 | m/s |
| Cruise speed | 9.8 | m/s |
| Stall speed | 8 | m/s |
| Design endurance | 6 | h |
| Maximum height | 6000 | m |

### 2.1. Energy System

The system topology of the UAV is shown in Figure 3, and it can be divided into energy and flight control systems. The flight control system is located above the power bus, which consumes power for autonomous control, and the energy system is located below the power bus, which is composed of

PV cells, maximum power point tracking (MPPT), and batteries for energy collection, consumption, and storage.

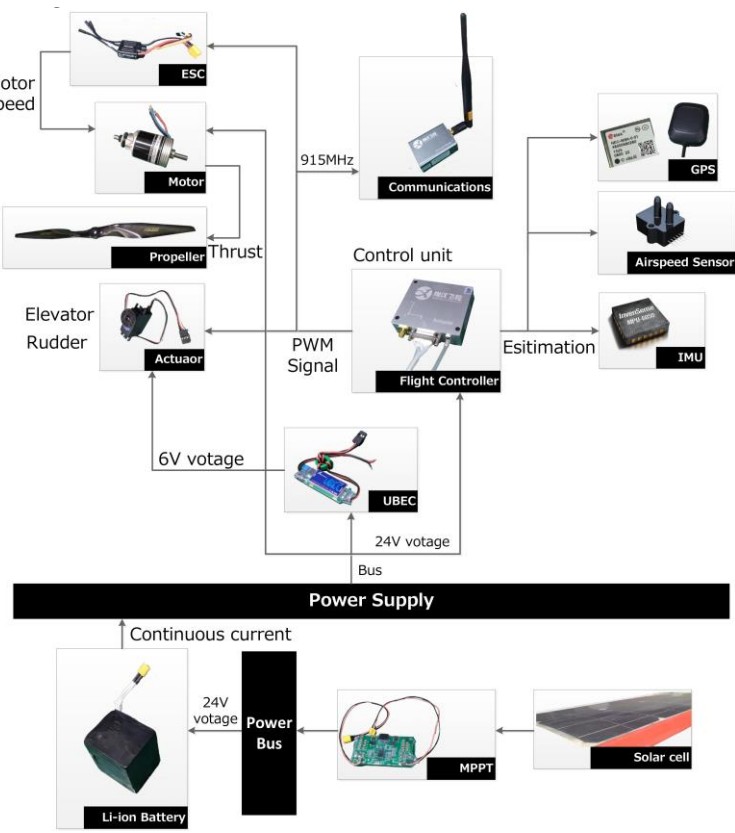

**Figure 3.** Topological system on overview of the solar UAV.

### 2.1.1. Solar Model

The solar model is based on the position, altitude, and attitude angle of the UAV to calculate the light intensity and the solar flux of the PV cells at the current time. Since the amount of solar energy changes with the latitude ($\xi$), longitude ($\lambda$), and altitude ($h$) of the earth, and the solar angles of azimuth ($\varphi_s$), elevation ($\varepsilon_s$) and Julian day ($j_d$) can be computed from ($\xi$, $\lambda$, $h$) and date/time [12]. The solar model is shown in Equation (1), which takes into account the annual variation and atmosphere absorption effects.

$$I = I_0\left(1 + 0.034 \cos \frac{2\pi j_d}{365}\right) f_a(h, \varepsilon_s) \tag{1}$$

where $I_0$ is the solar constant and $f_a(h, \varepsilon_s)$ is an atmosphere absorption factor [13].

The solar flux of the PV cells on the wing surface can be calculated based on the relationship between the vehicle coordinate frame, area of PV cells, and the location of the solar vector. In the Forward-Right-Down coordinate system, the solar flux area is given by

$$\Phi_k^s = A_k^s\begin{bmatrix} -\cos \varphi_s \cos \varepsilon_s & -\sin \varphi_s \sin \varepsilon_s & \sin \varepsilon_s \end{bmatrix} R_b^s \tag{2}$$

where $A_k^s$ is the area of each PV cells, $R_b^s = R(\psi)R(\theta)R(\varphi)$ is the rotation matrix form the body to the inertial frame, and $\psi$, $\theta$, and $\varphi$ are yaw, pitch, and roll angle.

### 2.1.2. Engine System

For the aileron-less control scheme, an accurate engine system model is helpful for the decrease of control input, and the engine system of the UAV is composed of a motor, a propeller, and an electronic

speed regulator (ESC). In this paper, the engine model is obtained based on the static and dynamic experiments [14], and the combination of DualSky-2814 brush-less motor and a 10-inch diameter, 7-inch pitch propeller is chosen according to the thrust requirement and efficiency on the standard operation condition. In the static experiment, the motor and propeller are mounted on an ATI Gamma 6-DOF force/torque sensor and then connected to a fixed clamp on the desk, and the static thrust and torque values are collected by the data acquisition hardware and software system. In the dynamic experiment, the thrust and torque are still measured, but the whole system needs to be fixed in the car, and the whole experiment system is shown in Figure 4.

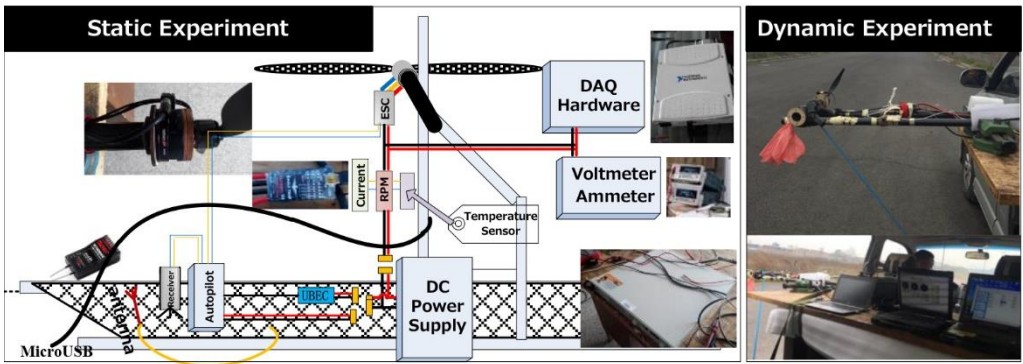

**Figure 4.** The setup of the propulsion system static and dynamic experiment.

The above experiment is carried out at a low altitude, but the UAV is intended to fly at high altitudes. Therefore, it was assumed that the propeller thrust and power coefficient did not change with altitude, and the calculation of high altitude propeller thrust and power was achieved by density change [15]. The nonlinear relationship between the static thrust and moment is shown in Figure 5a,b, and the dynamic thrust and moment are calculated based on the relationship between the advance ratio to the thrust and power coefficient, as shown in Figure 5c–e.

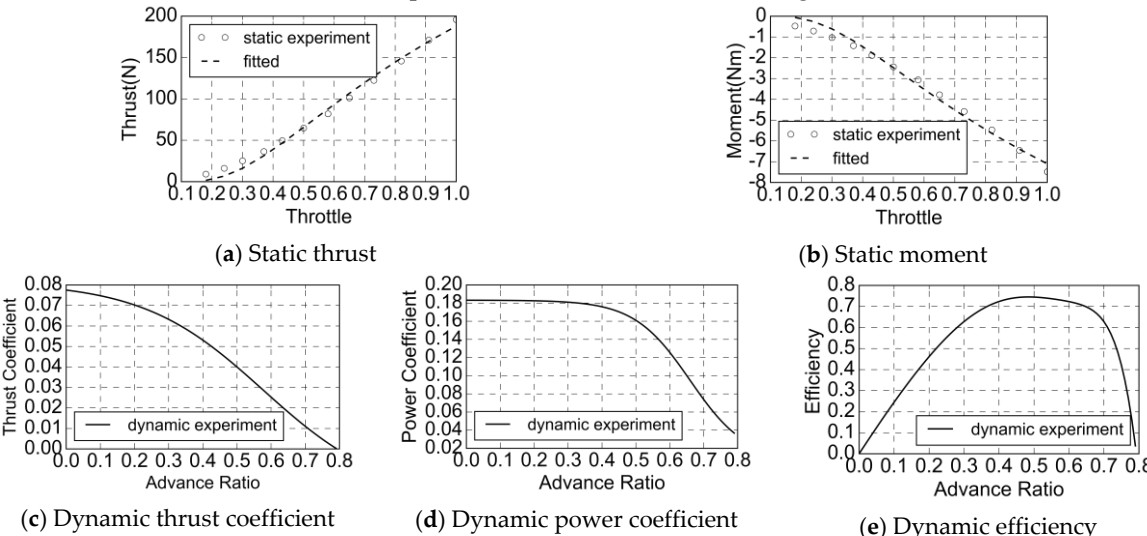

**Figure 5.** The static and dynamic experiment data.

The engine model is shown in Equation (3), where the thrust, power, and efficiency need to refer to the above experimental results.

$$J = \frac{n_s}{VD} \quad T_P = \rho n_s^2 D^4 C_T(J)$$
$$P_P = \rho n_s^3 D^5 C_P(J) \quad \eta_P = \eta(J) \quad M_P = P_P \cdot \eta_P / V \tag{3}$$

where $n_s$ is the motor speed, determined by the throttle; $J$ is the advance ratio; $\rho$ is the air density; $D$ is the propeller diameter; $V$ is the inflow speed, $V \approx V_a$; $T_P$, $P_P$, $\eta_P$, and $M_P$ represent the thrust, power, efficiency, and torque of propeller; and $C_T$ and $C_P$ represent the thrust and power coefficients.

### 2.1.3. Power and Energy Storage

The energy flow of the UAV is a typical structure of harvest–charge–discharge, where the absorbed solar energy is used as energy input to provide power for engine and accessory avionics systems, and the remaining energy is stored in batteries. When the batteries are full, the UAV will climb to store the gravity potential energy, and when the light intensity is insufficient, it will use the batteries as the only power input and slide to extend the flight endurance. The electric power input is given by

$$P_{in} = \eta_s \sum_k \Phi_k^s I \tag{4}$$

where $\eta_s$ is the solar collection system efficiency, $\eta_s = \eta_{panel} \cdot \eta_{MPPT}$, $\Phi_k^s I$ is the solar flux of each PV cell, and $\eta_{panel}$ and $\eta_{MPPT}$ are the efficiency of PV cell conversion and MPPT conversion, respectively. The collected energy can be stored in the battery and expended through the engine system or accessory avionics system for a stable flight. Power output is given by

$$P_{out} = P_{eng} + P_{acc} \tag{5}$$

where $P_{eng} = P_P/\eta_P$, $P_P$, and $\eta_P$ are shown in Equation (3), $P_{acc}$ is the accessory avionic power, including the payload, flight controller, data link, and actuators.

The battery model is designed according to three states: charge, hold, and discharge, with the efficiency of $\eta_{in}$ and $\eta_{out}$, and power flows into the battery system can be determined by the net power available ($P_{net} = P_{in} - P_{out}$), efficiency $\overline{\eta}$, and initial energy $E_{batt}(t_0)$ [16]; thus, the battery storage model is as follows:

$$E_{batt}(t) = \int_{t_0}^t \overline{\eta} P_{net} + E_{batt}(t_0)$$
$$\overline{\eta} = \begin{cases} \eta_{in} & P_{net} \geq 0 \cap E_{batt} < E_{max} \\ -1/\eta_{out} & P_{net} < 0 \cap E_{batt} > 0 \\ 0 & otherwise \end{cases} \tag{6}$$

where $E_{batt}$ is the stored energy in the batteries and $E_{max}$ is the maximum capacity. The energy system parameters are shown in Table 2.

**Table 2.** Energy system parameters. ESC: electronic speed regulator.

| Parameters | Value | Unit |
|---|---|---|
| Battery weight | 0.6 | kg |
| Battery energy density | 270 | Wh/kg |
| Battery capacity ($E_{max}$) | 168 | Wh |
| Surface area of solar cells | 0.65 | m$^2$ |
| Solar constant ($I_0$) | 1367 | W/m$^2$ |
| Accessory power ($P_{acc}$) | 15 | W |
| Motor efficiency ($\eta_{motor}$) | 0.95 | |
| ESC efficiency ($\eta_{ESC}$) | 0.97 | |
| $\eta_{panel}$ $\eta_{MPPT}$ $\eta_{in}$ $\eta_{out}$ | 0.20 0.92 0.93 0.95 | |

### 2.2. Flight Control System

The flight control system is installed at the center of gravity of the UAV, and its model consists of four parts: sensors, state estimation module, closed-loop control law, and actuator. Low-cost sensors are equipped in a flight controller for preliminary state measurement, with a hierarchical extended Kalman filter (EKF) algorithm applied to the state estimation module to achieve a reliable state for autopilot input. The control law is designed based on a successive loop closure structure;

then, the commands are output to the actuator to achieve autonomous control. Besides, the dynamic characteristics of the actuator are obtained by the experiment.

### 2.2.1. Sensors

The low-cost sensor combination of UAV includes the accelerometer, gyroscope, magnetometer, barometer, and GPS, and this sensor combination scheme has been widely used in small fixed-wing UAVs [17]. Different types of sensors have different measurement accuracy when high and low-frequency signals are input. For example, for high-frequency signals, the accelerometer has better measurement accuracy, but GPS has the opposite, and the yaw angle measured by the magnetometer is more reliable under windy conditions. Thus, the characteristics of this sensor combination on different signals can be used to achieve an optimal state estimation to improve control accuracy [11].

Low-cost sensors are susceptible to individual differences and external disturbances; consequently, bias, drift, and oscillatory need to be considered. Various sensors with the shelf product are shown in Table 3, with the SBG-Ellipse IMU component selected for high-precision comparison. The measurement signal includes traxial acceleration, angular velocity, magnetic angle, inertial position, ground speed, and static and dynamic pressure. The acceleration and angular velocity measurement undergo a two-layer function relationship between the voltage signal, displacement of the measuring element, actual acceleration, and Coriolis acceleration, as shown in Table 4. A triaxial magnetometer is composed of three monoaxial magnetic fields mounted orthogonally, in which combining the projection of geomagnetic direction on the body coordinate system gives the heading information [18]. The height and airspeed can be measured by static and dynamic pressure signals.

**Table 3.** Various sensors with measurement performance.

| Sensors | Dynamic Range (Horizontal Accuracy) | Nonlinear Degree (Accuracy/Update Rate) | Orthogonal Error (Error/Start Time) | Bias (Sensitivity) |
|---|---|---|---|---|
| MPU6050-Gyroscope | ± 2000 deg/sec | 0.2% | ± 2 | 0.05 deg/sec |
| Ellipse-Gyroscope | ± 450 deg/sec | 0.01% | 0.05 | 0.135 deg/sec |
| MPU6050-Accelerometer | ± 16 g | 0.5% | ± 2 | ± 50/± 50/±80 |
| Ellipse-Accelerometer | ± 8 g | 0.2% | 0.05 | - |
| LSM303D-Magnetometer | ± 12 gauss | 0.5% | 1 | ± 0.05 %/deg |
| MS5611-Barometer | 10–1200 mbar kPa | ± 1.5 mbar %Vfs | ± 2.5 mbar V/kpa | 0.5 ms |
| MPXV7002-Dynamic pressure | ± 2 mbar kPa | ± 2.5 mbar %Vfs | ± 1 mbar V/kpa | 1 ms |
| NEO-M8N-GPS | ± 2.5 m | 10 Hz | 26 s | −148 dbm |

**Table 4.** Measurement model of different sensors.

| Sensors | Measurement Model |
|---|---|
| Accelerometer | $\begin{cases} y_{accel,x} = \dot{u} + qw - rv + g\sin\theta + \beta_{accel,x} + \eta_{accel,x} \\ y_{accel,y} = \dot{v} + ru - pw + g\cos\theta\sin\varphi + \beta_{accel,y} + \eta_{accel,y} \\ y_{accel,z} = \dot{w} + pv - qu + g\cos\theta\sin\varphi + \beta_{accel,z} + \eta_{accel,z} \end{cases}$ |
| Gyroscope | $\gamma_{gyro} = k_{gyro}\Omega + \beta_{gyro} + \eta_{gyro}$ |
| Magnetometer | $y_{mag} = \psi + \beta_{mag} + \eta_{mag}$ <br> $B_0 = R^{-1}(\varphi, \theta, \psi)\left[0, 0, y_{mag}\right]^T$ |
| Barometer | $y_{abspres} = \rho g h_{AGL} + \beta_{abspres} + \eta_{abspres}$ |
| Airspeed sensor | $y_{diffpres} = \frac{\rho V_a^2}{2} + \beta_{diffpres} + \eta_{diffpres}$ |
| GPS | $y_{GPS_{n/e/h}}[n] = p_{n/e/h}[n] + v_{n/e/h}[n]$ <br> $V_{g_{GPS}} = \sqrt{(V_a\cos\psi + w_n)^2 + (V_a\sin\psi + w_e)^2} + \eta_V$ <br> $\chi_{GPS} = \tan^{-1}(V_a\cos\psi + w_n, V_a\sin\psi + w_e) + \eta_\chi$ |

In practical, the aforementioned sensors are often contaminated by amounts of measurement noise, which are assumed to be zero-mean Gaussian white noise. Meanwhile, the measurement bias is also included in the measurement process, especially if there is mutual interference and temperature variation. The measurement models of different sensors are shown in Table 4.

Here, $y$ represents the measured data, $\Omega = [p, q, r]^T$ is the angular velocity, $\beta$ is a temperature-related bias, and $\eta$ is the Gaussian white noise, $\eta \sim N(0, Q)$. For the GPS measurement shown in Table 4, the position error and its dynamic characteristics are both required, and a Gauss–Markov error model is an appropriate description of the GPS measurement process [19].

$$v[n+1] = e^{-k_{GPS}T_s}v[n] + \eta_{GPS}[n] \tag{7}$$

According to the flight conditions of the UAV at low altitude, with an altitude of 600 m and an airspeed of 12.5 m/s during the hovering phase, the sensors are simulated with no wind, an altitude of 600 m, a turning radius of 80 m, the command rolling angle of 20 degrees; the controller temperature is 25 °C and remains unchanged, combined with the orthogonal error and bias from Table 3. The simulation results of various low-cost sensors are shown in Figure 6, the *meas-sim* and *true-sim* represent the measurement state and real state of the sensor simulation, respectively.

Figure 6a,b, which is the Inertial Measurement Unit (IMU) simulation, where the accelerometer measurement is a high-frequency signal and the noise of low-cost sensors is obvious; (c) is a magnetometer, the measurement is a medium-low-frequency signal; (d,e) is GPS; and (f) is the pressure sensors' simulation, as low frequency as GPS, refer to Table 4. In the dynamic process, the measurement is consistent with the trend of the true signal, and when entering the steady state, the measurement noise is large, and there is a significant deviation.

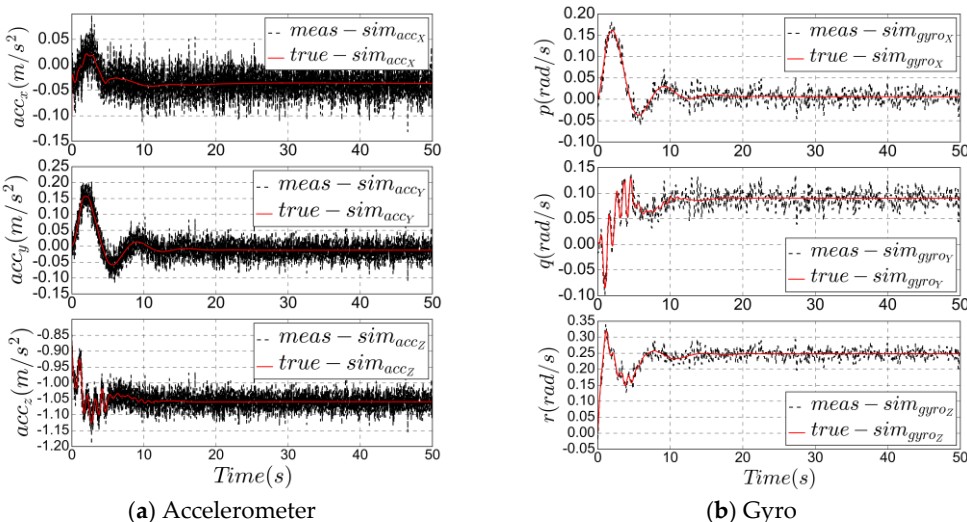

(**a**) Accelerometer      (**b**) Gyro

**Figure 6.** *Cont.*

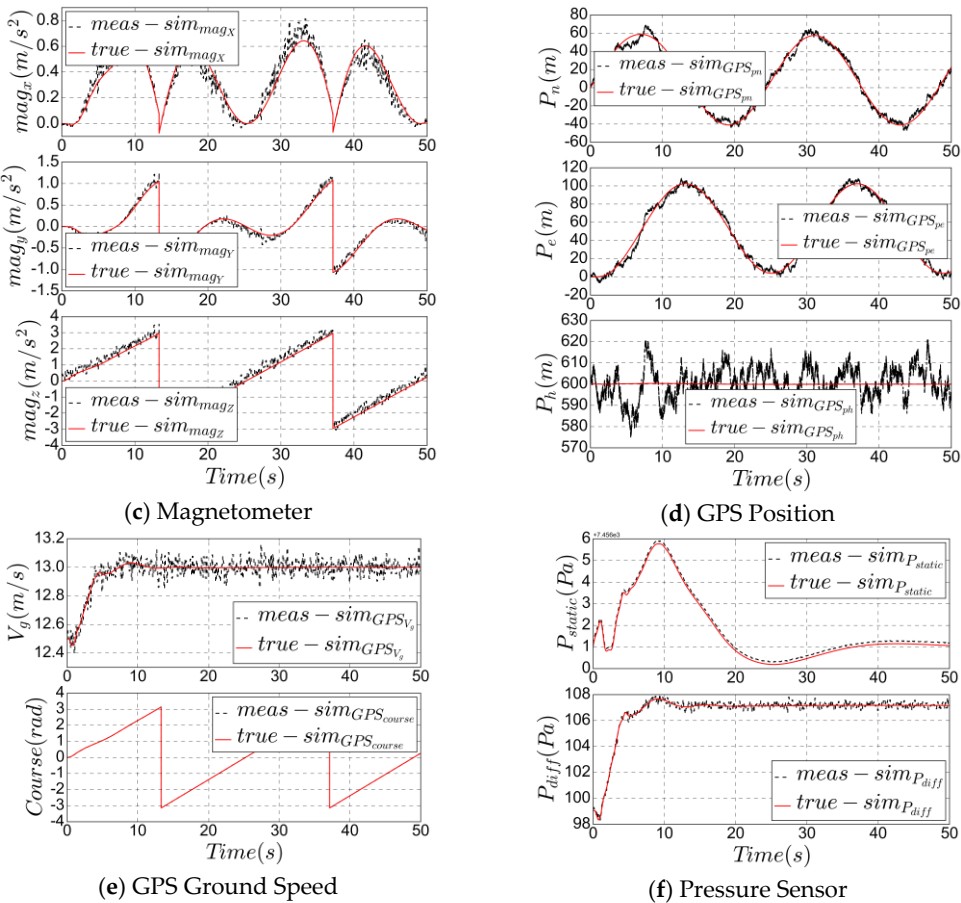

(**c**) Magnetometer          (**d**) GPS Position

(**e**) GPS Ground Speed         (**f**) Pressure Sensor

**Figure 6.** True states and measurement states comparison.

### 2.2.2. State Estimation

After obtaining the measurement data of the sensor, a dynamic observation of the raw data is necessary for low-cost hardware. Extended Kalman Filter (EKF) theory is a suitable method for the poor computing power of a low-cost flight controller, which has been widely used in various open-source flight controllers [20], and different EKF structures are derived according to different sensor combinations; for example, there is no magnetometer or more angle of attack and sideslip angle sensors [21]. The UAV adopts the EKF structure algorithm for state estimation, which consists of prediction and update steps, and the state equation and measurement equation for nonlinear discrete dynamical systems can be written as

$$
\begin{aligned}
\dot{\mathbf{x}} &= \mathbf{f}(\mathbf{x}, \mathbf{u}) + \mathbf{w}(t) \\
\mathbf{z}_k &= \mathbf{h}(\mathbf{x}(t_k), \mathbf{u}) + \mathbf{v}_k
\end{aligned}
\tag{8}
$$

where $\mathbf{w}(t)$ represents the noise and unmodeled part, which can be considered as a zero-mean Gaussian noise with covariance $Q$. The random variable $\mathbf{v}_k$ is the measurement noise and represents noise on the sensors with covariance $R$, which can be estimated from sensor calibration, but $Q$ is generally unknown and needs to be tuned in order to improve the performance of the observer; thus, the prediction step is given by

$$
\begin{aligned}
\dot{\hat{\mathbf{x}}} &= \mathbf{f}(\hat{\mathbf{x}}, \mathbf{u}) \\
\mathbf{A}(\hat{\mathbf{x}}, \mathbf{u}) &= \frac{\partial \mathbf{f}(\mathbf{x}, \mathbf{u})}{\partial \mathbf{x}}|_{\mathbf{x}=\hat{\mathbf{x}}} \\
\dot{\mathbf{P}} &= \mathbf{A}(\hat{\mathbf{x}}, \mathbf{u})\mathbf{P} + \mathbf{P}\mathbf{A}(\hat{\mathbf{x}}, \mathbf{u})^T + \mathbf{Q}.
\end{aligned}
\tag{9}
$$

The update step is

$$
\begin{aligned}
\mathbf{C}(\mathbf{x}, \mathbf{u}) &= \frac{\partial \mathbf{h}(\mathbf{x}^-, \mathbf{u})}{\partial \mathbf{x}} \\
\mathbf{L} &= \mathbf{P}^- \mathbf{C}^T \left( \mathbf{R} + \mathbf{C}(\hat{\mathbf{x}}, \mathbf{u}) \mathbf{P} \mathbf{C}(\hat{\mathbf{x}}, \mathbf{u})^T \right)^{-1} \\
\mathbf{P} &= (\mathbf{I} - \mathbf{L} \mathbf{C}(\hat{\mathbf{x}}, \mathbf{u})) \mathbf{P}^- \\
\hat{\mathbf{x}} &= \hat{\mathbf{x}}^- + \mathbf{L}(\mathbf{z} - \mathbf{h}(\hat{\mathbf{x}}^-, \mathbf{u}))
\end{aligned}
\tag{10}
$$

where $\mathbf{A}$ is a linearized state update matrix, $\mathbf{C}$ is a linearized model output matrix, $\mathbf{P}$ is the state covariance matrix, and $\mathbf{L}$ is the sensor update gain matrix.

Based on the EKF theory, a three-stage series structure algorithm is used in the state estimation module, as shown in Figure 7. The estimation of pitch and roll in the first stage is used as an input to the second stage to calculate the heading angle. The information of the first two stages along with GPS measurement is used as the input to the final stage for the position and wind estimation. During the flight, the heading state such as yaw and heading angle will directly affect the trajectory tracking accuracy, while the longitudinal state such as airspeed and altitude have less influence, since the installation of a PV cell on the wing will interfere with the measurement of the magnetometer and reduce the accuracy of the yaw angle. The key of this algorithm is to reduce the airspeed and altitude estimation accuracy, improve the heading angle accuracy, and make full use of the limited computing ability to improve the position tracking precision to meet the mission requirement. For example, a four-sided route with an area of 1 km$^2$ has an acceptable tracking error of approximately 30 m (30 m/km$^2$) and a heading error of nearly 13 degrees, and the simulation conditions are the same as before with an altitude of 600 m, an airspeed of 12.5 m/s, and a rolling angle hovering of 20 degrees in windless conditions [10].

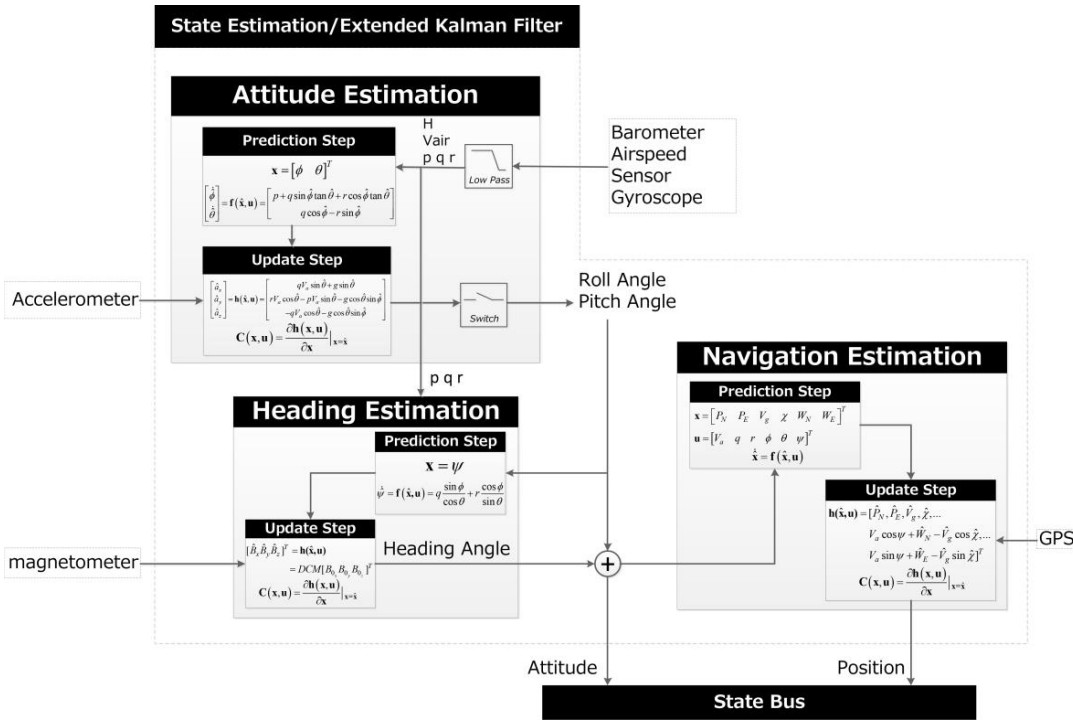

**Figure 7.** Architecture of the three-stage state estimation framework.

### 2.2.3. Successive Loop Closure Control Law

For the low-cost solar UAV, the flight control architecture emphasizes simplicity, robustness, and low-power consumption to fulfill the need for a reliable long-endurance automatic flight. Thus, the structure of the control law is designed by using the longitudinal and lateral separated cascaded proportional–integral–derivative (PID) controllers, as shown in Figure 8. In the outer loop, the flight

controller employs a nonlinear guidance law to track waypoints by generating the command roll angle. Altitude and airspeed control are provided by a proportional–integral (PI) structure, command pitch angle, and throttle as output [22].

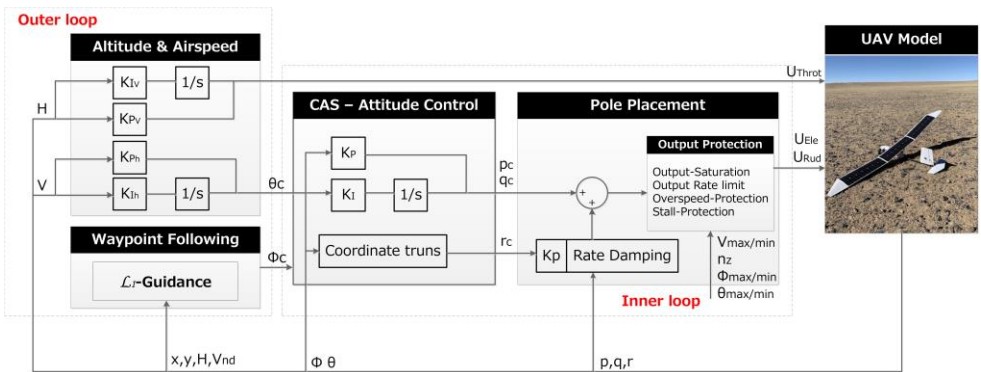

**Figure 8.** Cascaded control scheme for solar-powered UAV.

The target application field of the UAV is a plateau with an altitude of 5 km and a cruise speed of 12 m/s, and the flight modes under trim conditions are shown in Table 5, which are all stable. The longitudinal control inputs are throttle and elevator, and the lateral input is only the rudder, so a pseudo aileron is defined here. Since the longitudinal control law adopts a conventional control structure, only the lateral control law is studied here, which is designed according to the combination of roll attitude control and coordinated turning control.

**Table 5.** Basic flight modes of the longitudinal and lateral system dynamics.

| Mode | Eigenvalue | Nature Frequency (rad/s) | Damping Ratio |
|------|-----------|--------------------------|---------------|
| Longitudinal | | | |
| Phugoid | −8.10 ± 5.71i | 0.839 | 0.046 |
| Short Period | −0.039 ± 0.84i | 9.912 | 0.817 |
| Lateral | | | |
| Spiral | −0.53 | - | - |
| Dutch Roll | −0.28 ± 2.26i | 2.277 | 0.122 |
| Roll | −13.14 | - | - |

For the pseudo aileron and rudder channels, the control law of lateral inner loop is as follows

$$\begin{cases} \delta_{a0} = (k_p + k_{p_i} \cdot 1/s + k_{p_d} \cdot s)(p_c - p)p_c = k_\phi(\phi_c - \phi) \\ \delta_{r0} = k_r(r_c - r)r_c = g/V_0 \tan\phi_c \\ \delta_r = k_{\delta_a} \cdot \delta_{a0} + k_{\delta_r} \cdot \delta_{r0} \\ k_{\delta_a} = \text{sat}\{0, |p|/|r|, 1\}k_{\delta_a} + k_{\delta_r} = 1 \end{cases} \tag{11}$$

where $\delta_{a0}$ is the pseudo aileron, $\delta_{r0}$ is the pseudo rudder, and $k_{\delta a}$ and $k_{\delta r}$ are the proportion of aileron and rudder in the channel, which are related to the roll angular velocity $|p|$ and yaw angular velocity $|r|$, respectively. There are five circumstances for rudder control structure design, as shown in Table 6 and Figure 9a. Among them, the roll angle hold can be achieved by any control method, but the best control effect is the combination of coordinate turning and PI control of aileron, and the worst is obtained by directly inputting the aileron command to the rudder, with the best control law is loaded into the flight

controller hardware. In the lateral out loop, $L_1$ guidance law [23] is applied for trajectory following, as shown in Equation (12).

$$a_{cmd} = g \tan \varphi = \frac{V^2}{R} \approx \frac{V}{L_1}\left(\dot{d} + \frac{V}{L_1}d\right)$$
$$\varphi_{cmd} = \arctan\left(\frac{V}{L_1 g}\left(\dot{d} + \frac{V}{L_1}d\right)\right)$$

(12)

**Table 6.** Summary of UAV design and performance characteristics.

| Number | Control Structure | Response Time (s) | Steady-State Error | Ranking |
|:---:|:---:|:---:|:---:|:---:|
| 1 | Coordinated turn | 3 | 2% | 3 |
| 2 | Coordinated turn and (P) aileron control | 3 | 2% | 3 |
| 3 | Coordinated turn and (PI) aileron control | 2.4 | 0 | 1 |
| 4 | Coordinated turn and (PID) aileron control | 2.875 | 0 | 2 |
| 5 | (PID) aileron control | 4.7 | 2.4% | 5 |

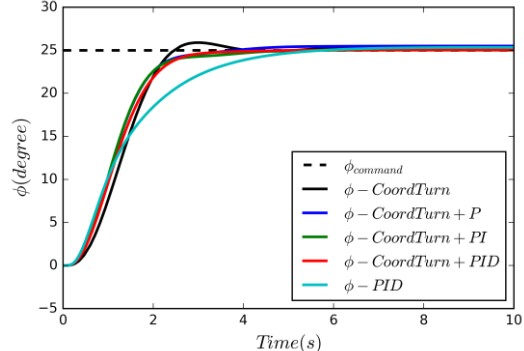

(**a**) Roll angle control under different control modes

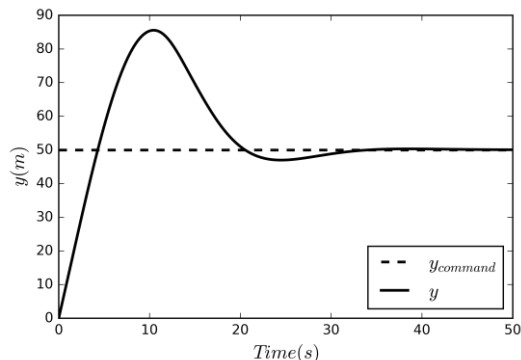

(**b**) Straight-line trajectory tracking

**Figure 9.** Roll angle hold comparison and straight-line path tracking.

The five roll attitude hold simulations comparison and straight-line path following are shown in Figure 9, and the UAV starts with a given trim state, in which the roll angle command is 25 degrees and the command path is from (0, 50) to (1000, 50). The roll angle command can be quickly tracking due to the absence of direct control of the ailerons, while the lateral trajectory tracking response is slow, but it still can be used in practice.

## 2.2.4. Actuator

The UAV adopts a digital actuator as the only driving mechanism of its aerodynamic surface to generate control force and moment, since the response time of the actuator is generally much smaller than its time constant. In the initial stage of the controller design, the influence of the actuator can be ignored. However, the dynamic response of the actuator will affect the actual control process in the flight test stage, and the slow response of the servo may cause the entire system to diverge, so the response characteristics of actuator must be considered [24]. In the simulation, the dynamic process of a digital actuator can be regarded as a linear second-order model, with the effects of the dead zone, saturation, clearance, and communication delay [25], and the damping and frequency characteristics are still need to be measured experimentally. The dynamic performance test experiment of the actuator is completed in this paper, as shown in Figure 10, and a sweep frequency function is input through the Nano system.

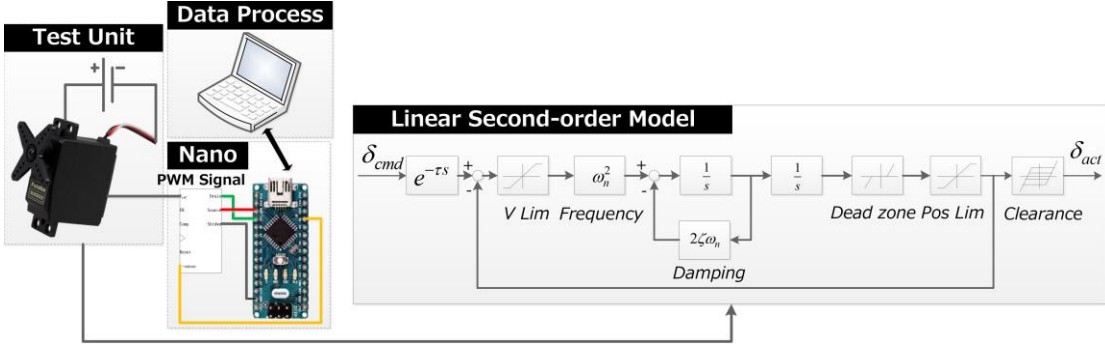

**Figure 10.** Actuator dynamic performance test experiment scheme.

During the experiment process, the actuator receives a 50 Hz PWM signal as an input, and the test time is 50 seconds. The test command is $\delta_{cmd} = \delta_{\max} \sin\left(f_0 t + kt^2/2\right)$, in which $\delta_{max}$ is 20 degrees, $f_0$ is 1, and $k$ equals to 0.2, and the transfer function of the actuator can be expressed in Equation (13).

$$\Phi(s) = \frac{9937}{s^2 + 332.3s + 10660}e^{-0.021s} = \frac{9937}{(s + 36.0)(s + 296.3)}e^{-0.021s} \tag{13}$$

## 3. Simulation Verification

Before implementing the control system on the UAV, the whole system needs to be verified by simulations, including a numerical nonlinear model and hardware-in-the-loop (HIL) validation. The numerical simulation is used to justify the whole platform, aerodynamic, energy, and control system, and to test each phase and state-switching process via a typical mission route [26]. After the numerical simulation verification, combined with the control system hardware and a 6-DOF motion platform, an HILS system is established for typical flight modes, sensor measurement, and control logic.

### 3.1. Numerical Simulation

In the numerical simulation, the conditions are set according to the plateau environment conditions during flight test, a typical mission route was used, and the flight time was December 21, 2018, at 10:57, with an altitude of 5000 m, a mission height of 100 m above the ground level, a temperature of 0 °C, and no wind. The initial capacity of the battery was 70% of the maximum capacity, and the estimated state was used as feedback. The simulation results of the flight path, position, and attitude of different phases are shown in Figure 11, where the estimated state is indicated with a dashed line, and the true state is indicated with a solid line for comparison.

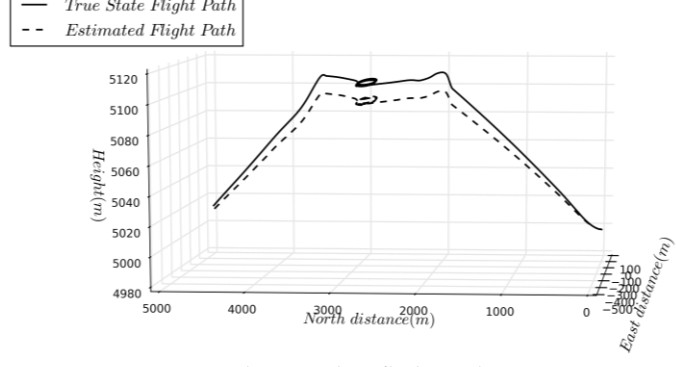

(**a**) The complete flight path

**Figure 11.** *Cont.*

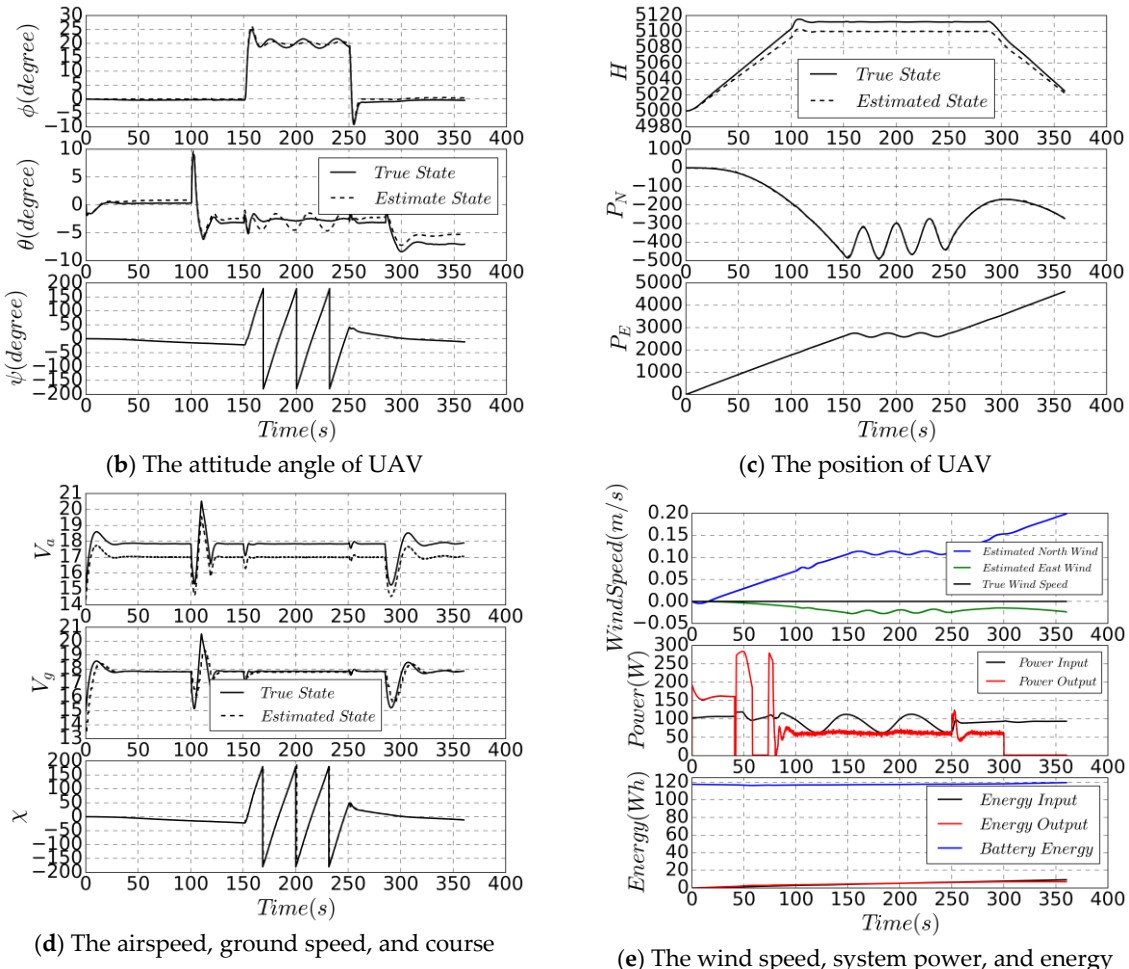

(**b**) The attitude angle of UAV

(**c**) The position of UAV

(**d**) The airspeed, ground speed, and course

(**e**) The wind speed, system power, and energy

**Figure 11.** Complete mission process simulation.

Figure 11a shows a comparison between the estimated and true flight trajectories. Since the altitude and airspeed are only low-pass filtered, the static bias of the altitude is 12 m, and the airspeed bias is 1 m/s, as shown in Figure 11c,d. Figure 11b shows the change of the attitude angle, 150–250 s is the phase of the hovering mission, the roll angle error range is 3 degrees, and the pitch angle is 2 degrees, which makes the hovering trajectory fluctuate slightly, but the trajectory tracking process remains stable. Figure 11d,e shows that 8 m is a reliable estimation of the heading angle, ground speed, and wind speed ensuring the low-cost platform has a sufficient position accuracy with the trajectory tracking accuracy. The above results show that the accuracy of the longitudinal parameters has less effect on the mission trajectory, while the attitude and heading will have a direct impact, and the attitude angle changes obviously when the state is switched, such as climbing transfer to cruise state, but the position parameters such as ground speed and heading angle remain stable. Besides, the accuracy of the estimated states is shown in Table 7 with the precision sequence in the reverse of the state estimation stages i.e., "position > wind speed > heading/ground speed > attitude > altitude/airspeed". In addition, as shown in Figure 11e, the input energy is slightly greater than the output, the battery is in a balance of charging and discharging, the maximum output power during the climb is 280 W, the cruise power is 64 W, and the energy output will increase obviously during the state-switching process. Therefore, it is possible to increase endurance by reducing state switching in a complete mission.

**Table 7.** The error precision of flight parameters in different phases.

| Phase | Position (m) | Attitude (deg) | Heading (deg) | Ground Speed (m/s) | Airspeed (m/s) | Height (m) | Wind (m/s) |
|---|---|---|---|---|---|---|---|
| Climb | 3.5 | 0.59 | 0.26 | 0.31 | 0.89 | 11.07 | 0.6 |
| Descend | 4.5 | 0.87 | 0.57 | 0.44 | 0.88 | 11.94 | 0.19 |
| Cruise | 6.4 | 0.81 | 0.22 | 0.53 | 0.98 | 12.37 | 0.4 |
| Hovering | 7.9 | 1.82 | 0.4 | 0.38 | 0.86 | 12.03 | 0.7 |

## 3.2. Stewart Platform (SP) Modeling

HILS is an effective way for the implementation of advanced control law to hardware, which bridges the practical real-time control and the numerical simulation together [27]. Compared with the three-axis motion platform, SP is more open and has a greater advantage in load capacity and space, the UAV can be directly fixed to the platform for a complete HILS experiment, and the platform can perform 6-DOF motions to realize the coupling of linear and angular motion. In this paper, three experimental motion modes are proposed including attitude, angular velocity, and acceleration to test different dynamic characteristics. The Jiwang Mechatronics Company (JWMC)'s motion platform and simulation model used here are shown in Figure 12, and its motion limitations are shown in Table 8.

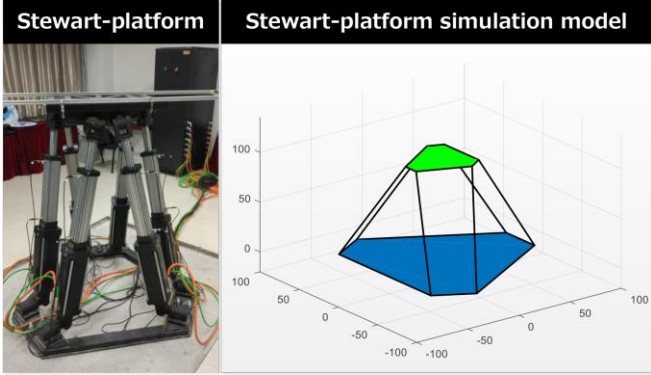

**Figure 12.** Stewart platform (SP) physical and simulation model.

**Table 8.** Limit parameters of SP in xyz direction.

| | X Angular | Y Angular | Z Angular | X Linear | Y Linear | Z Linear |
|---|---|---|---|---|---|---|
| Attitude/Position | 20° | 20° | 30° | 100 mm | 100 mm | 80 mm |
| Velocity | 70°/s | 70°/s | 80°/s | 1000 mm/s | 1000 mm/s | 1000 mm/s |
| Acceleration | 200°/s$^2$ | 200°/s$^2$ | 200°/s$^2$ | 2000 mm/s$^2$ | 2000 mm/s$^2$ | 1000 mm/s$^2$ |

The SP is limited by its range of motion and cannot simulate feature points in the whole flight envelope, but it can realize the specified frequency and amplitude motion within the limitation, such as short-period or phugoid motion, or specific dynamic processes, such as individual velocity or acceleration simulation. Therefore, the movement of the SP needs to be modeled to predict the attitude and position of the load platform and to verify the expression of the platform for the tested motion.

Figure 13 shows the geometry of SP, in which the base platform is fixed to the ground, and the load platform moves according to the length of the six hydraulic struts [28]. Here, we define {P} and {B} as the coordinate systems of the load and the base platform, and $O_p$ and $O_b$ as the origins of these two coordinate systems. Thus, the base platform is fixed and the points on the load platform can be represented as vectors in this coordinate system. Let **t** be a vector from $O_b$ to $O_p$, **t** = $(x\ y\ z)^T$ and **θ** be a Cartesian angle vector of {P} relative to {B}, **θ** = $(\alpha\ \beta\ \gamma)^T$, assuming that the vector from the connection point of the load platform to $O_p$ is **$p_i$**, and the vector from the connection point of the base platform to $O_b$ is **$q_i$**.

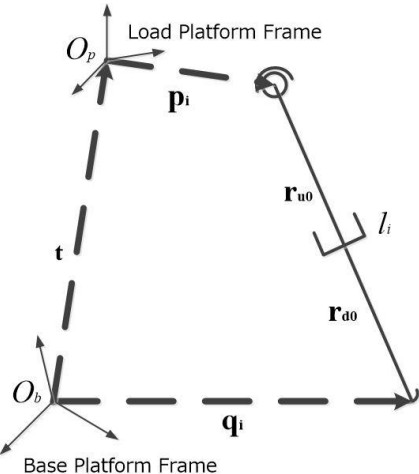

**Figure 13.** Kinematic relationship of Stewart platform.

The vector of the connection point from the base to the load platform is $\mathbf{S_i}$, which is represented as

$$\mathbf{S_i} = \mathbf{Rp_i} + \mathbf{t} - \mathbf{q_i}. \tag{14}$$

The hydraulic strut length and unit vector are

$$
\begin{aligned}
L_i &= \|\mathbf{S_i}\| \\
\mathbf{s} &= \mathbf{S_i}/L_i = (\mathbf{Rp_i} + \mathbf{t} - \mathbf{q_i})/\|\mathbf{Rp_i} + \mathbf{t} - \mathbf{q_i}\|
\end{aligned}
\tag{15}
$$

Since $\mathbf{L_i}$ is a function of $(x, y, z, \alpha, \beta, \gamma)$, with $\boldsymbol{\omega} = \dot{\theta}$, $\mathbf{v} = \dot{\mathbf{t}}$, $\dot{\chi} = \left(\mathbf{v}^T, \boldsymbol{\omega}^T\right)$, $\mathbf{q_i} = \mathbf{Rp_i}$, the slide speed of hydraulic strut is

$$\mathbf{S_i} = \mathbf{s_i} \times (\mathbf{v} + \boldsymbol{\omega} \times \mathbf{q_i}). \tag{16}$$

Equation (16) can be further derived as

$$
\begin{aligned}
\mathbf{S}_i &= \mathbf{s}_i \times (\mathbf{v} + \boldsymbol{\omega} \times \mathbf{q}_i) \\
&= \mathbf{s}_i^T \mathbf{v} + \mathbf{s}_i \cdot (\boldsymbol{\omega} \times \mathbf{q}_i) = \mathbf{s}_i^T \mathbf{v} + (\boldsymbol{\omega} \times \mathbf{q}_i)^T \\
&= \left(\begin{array}{cc} \mathbf{s}_i^T & (\mathbf{q}_i \times \mathbf{s}_i)^T \end{array}\right)\left(\begin{array}{c} \mathbf{v} \\ \boldsymbol{\omega} \end{array}\right) = \left(\begin{array}{cc} \mathbf{s}_i^T & -\mathbf{s}_i^T \mathbf{q}_i^\times \end{array}\right)\left(\begin{array}{c} \mathbf{v} \\ \boldsymbol{\omega} \end{array}\right) \\
&= \left(\begin{array}{cc} \mathbf{s}_i^T & -\mathbf{s}_i^T \tilde{\mathbf{q}}_i \end{array}\right)\left(\begin{array}{c} \mathbf{v} \\ \boldsymbol{\omega} \end{array}\right) = \mathbf{J}\dot{\chi}
\end{aligned}
\tag{17}
$$

where the Jacobian matrix can be expressed as

$$\mathbf{J} = \left(\begin{array}{cc} \mathbf{s_i}^T & (\mathbf{q_i} \times \mathbf{s_i})^T \end{array}\right). \tag{18}$$

### 3.3. Stewart Platform Hardware-In-the-Loop-Simulation (SPHILS)

The SPHILS system consists of an SP, upper machine for simulation preset, lower machine for model calculation, industrial computer for motion platform drive, in which the flight controller receives the measurement information output control command, and the ground station receives real-time information and sends instructions. This co-simulation system can be used to verify the feasibility of the control system, calibrate the sensor and filter algorithm, and reproduce the typical flight mode. Figure 14 shows the system structure of the SPHILS, which can be divided into three layers: simulation environment, hardware, and output. The real-time simulation environment is built in MATLAB/Simulink, and the upper machine is embedded with a complete system including a controller model, nonlinear dynamics, desktop real-time simulation module, and the SP model. The

hardware layer consists of a flight controller and SP, and it communicates with the simulation system through RJ45 and COM ports to express motion. The ground station and external servos are the output layer and obtain the measurement parameters and control commands.

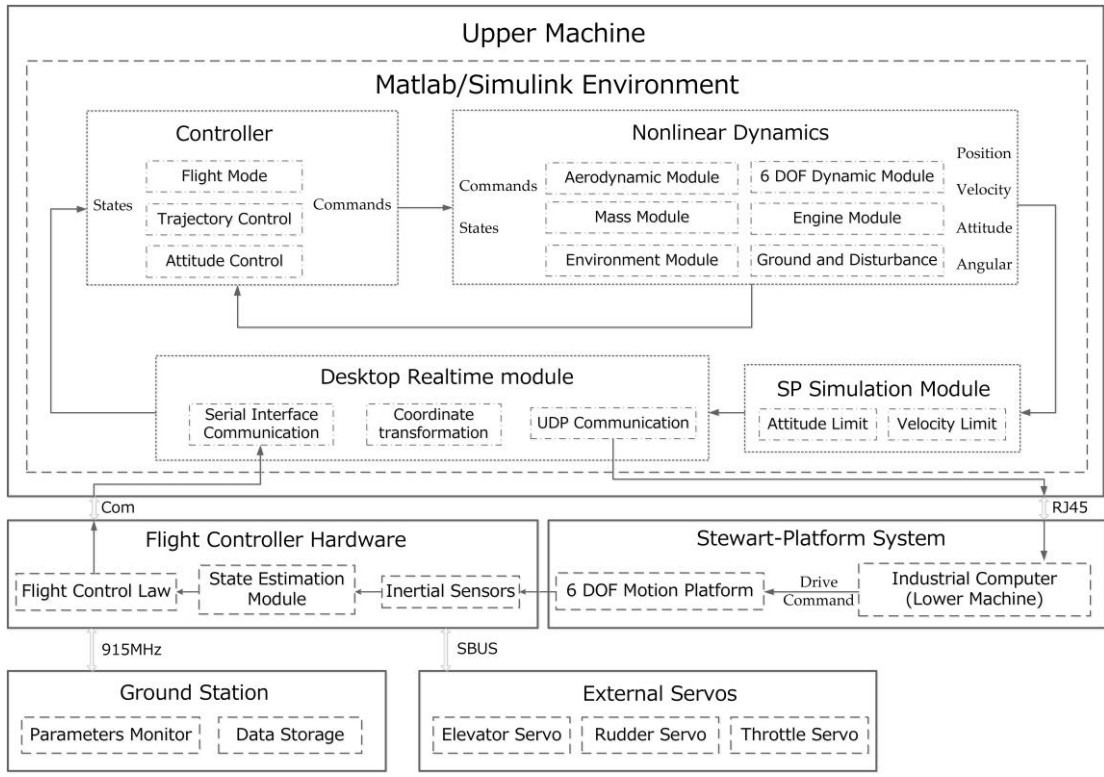

**Figure 14.** Schematic diagram of the UAV hardware-in-the-loop-simulation (HILS).

Figure 15 shows an open-loop and a closed-loop experimental scheme of the real HILS system. The open-loop experiment is a process from the upper computer to the ground station, which is a unidirectional structure and tests the limit states in the process of simulation, such as acceleration and angular velocity exceeding the limit, and it also verifies the consistency of the instructions between the flight controller and simulation model. The closed-loop experiment is a structure of feedback, in which the real measured value of the sensor is input into the simulation model of the upper machine instead of the simulation model of sensor; then, the sensor measurement and the state estimation algorithm of the dynamic process can be verified.

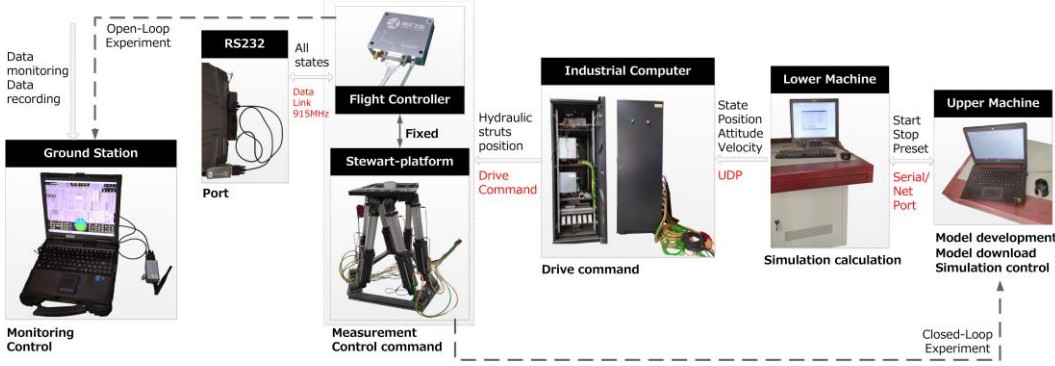

**Figure 15.** Real-time HIL simulation environment.

The attitude angle and position in the simulation model are relative to the aircraft body coordinate system $Ox_by_bz_b$, which is different from the platform's own coordinate system $Ox_Sy_Sz_S$, and it is shown as follows.

As shown in Figure 16, the conversion relationship between the SP coordinate system and UAV body coordinate system can be obtained as follows:

$$\begin{bmatrix} x_S \\ y_S \\ z_S \end{bmatrix} = \begin{bmatrix} 0 & -1 & 0 \\ 0 & 0 & -1 \\ 1 & 0 & 0 \end{bmatrix} \begin{bmatrix} x_b \\ y_b \\ z_b \end{bmatrix}. \tag{19}$$

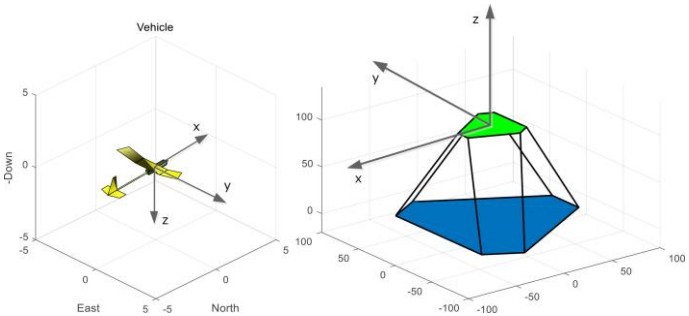

(**a**) UAV simulation model      (**b**) SP simulation model

**Figure 16.** The coordinate system of SP and UAV.

*3.4. Experiment Design*

In order to verify the stable measurement and control of the flight controller in the complete envelope, the above SPHILS system is used for control system experiment. Although the range and velocity are limited, the movement can be decomposed to a single direction to ensure that the test parameters do not exceed the platform limitation. In the paper, the parameters of aircraft are classified according to the response speed; for example, the acceleration is the most direct, the next is the linear or angular velocity, and the last is the position and attitude. Thus, three experiment modes are proposed including acceleration, angular velocity, and attitude. The acceleration mode focuses on the load simulation, the angular velocity mode is for each flight mode, and the attitude mode is a single-point test to calibrate the measurement of the controller.

3.4.1. Acceleration Mode

In the acceleration mode experiment, only a single acceleration movement in a certain direction is carried out to verify the limit overload of the aircraft and calibrate the measurement of angular velocity in the dynamic process. When the aircraft is fixed on the platform, this mode can be used to test the limit frequency and amplitude of structural strength. Figure 17 shows the test results under 50 and 100°/s$^2$ angular acceleration, in which the solid line is the input of the upper machine, and the dotted line is the measurement of the flight controller. The delay time of the open-loop experiment is 100 ms, and the closed loop is increased to 500 ms due to the feedback of the controller hardware; so, the acceleration mode requires the system to have a fast response and can only be tested in the open-loop experiment. Results show that the control system can withstand a 100°/s$^2$ vibration, and the pitch angle and angular velocity can be accurately measured. The movement in this mode is gradually divergent, and it can be regarded as an unstable state in practice. In addition, the whole process of oscillation divergence can be reproduced by the experiment.

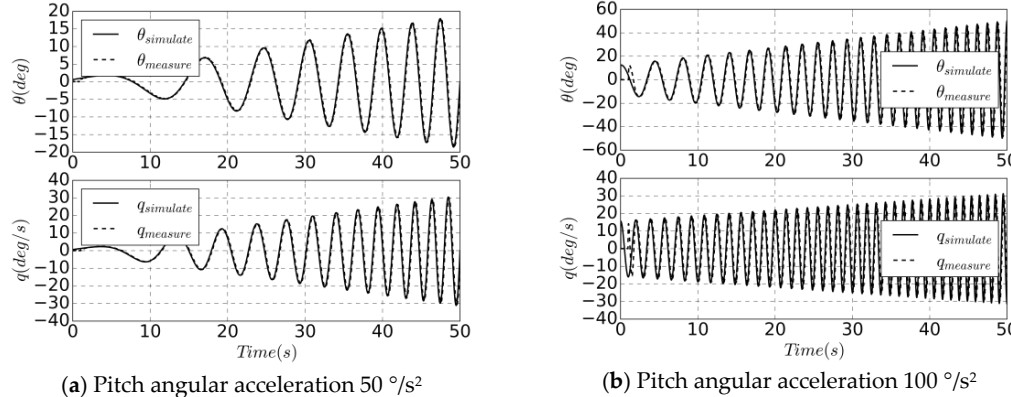

(**a**) Pitch angular acceleration 50 °/s²

(**b**) Pitch angular acceleration 100 °/s²

**Figure 17.** Test results under different angular accelerations.

### 3.4.2. Angular Velocity Mode

The angular velocity mode is a fixed frequency and amplitude simulation, and it can be used for both open-loop and closed-loop experiments of typical flight modes as shown in Table 5, as well as a calibration of angular velocity measurement and attitude estimation. In this paper, the frequencies of 0.15 and 0.012 Hz are used as short period and phugoid for open-loop and closed-loop experiments, as shown in Figure 18.

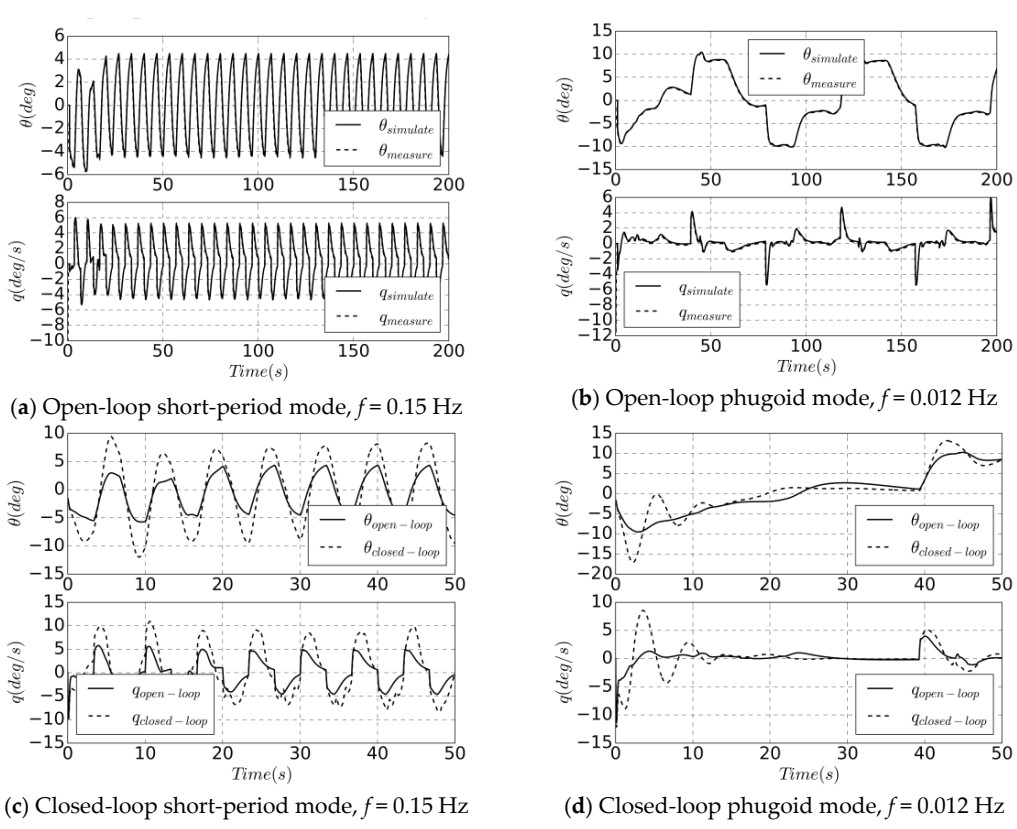

(**a**) Open-loop short-period mode, $f$ = 0.15 Hz

(**b**) Open-loop phugoid mode, $f$ = 0.012 Hz

(**c**) Closed-loop short-period mode, $f$ = 0.15 Hz

(**d**) Closed-loop phugoid mode, $f$ = 0.012 Hz

**Figure 18.** HILS open-loop and closed-loop experiment results.

Figure 18a,b shows that SP can achieve an accurate tracking of pitch angle; there is a small error in pitch angular velocity tracking, and the attitude angle input is consistent with the measurement. In the closed loop as shown in Figure 18c, as the measurement state feedback to the upper machine, the amount of calculation becomes larger, and the motion exhibits a certain delay. The experiment frequency of the closed loop is the same as that of the input, but the amplitude increases by nearly

30%. Combined with Figure 18d, the closed-loop experiment also shows that the amplitude increases obviously in the state change process. Compared with the numerical simulation, the dynamic process of the state change in practice will have a certain delay, and the amplitude will be enlarged or reduced. By adjusting the dynamic characteristics of the SPHIL, such as adding a given delay process or amplitude limit to the acceleration or velocity of the platform, it can reproduce the actual motion characteristics and reduce the gap between the numerical simulation and the real flight.

### 3.4.3. Attitude Mode

The attitude mode is a simulation based on the state points in the whole flight process, as shown in Figure 19, and there are four constant states in different missions. Compared with the first two experiments, the attitude mode is static and can be used to verify the control logic of each point. By comparing the real output of the controller with the model output, the gain parameters and measurement accuracy of the controller are further calibrated.

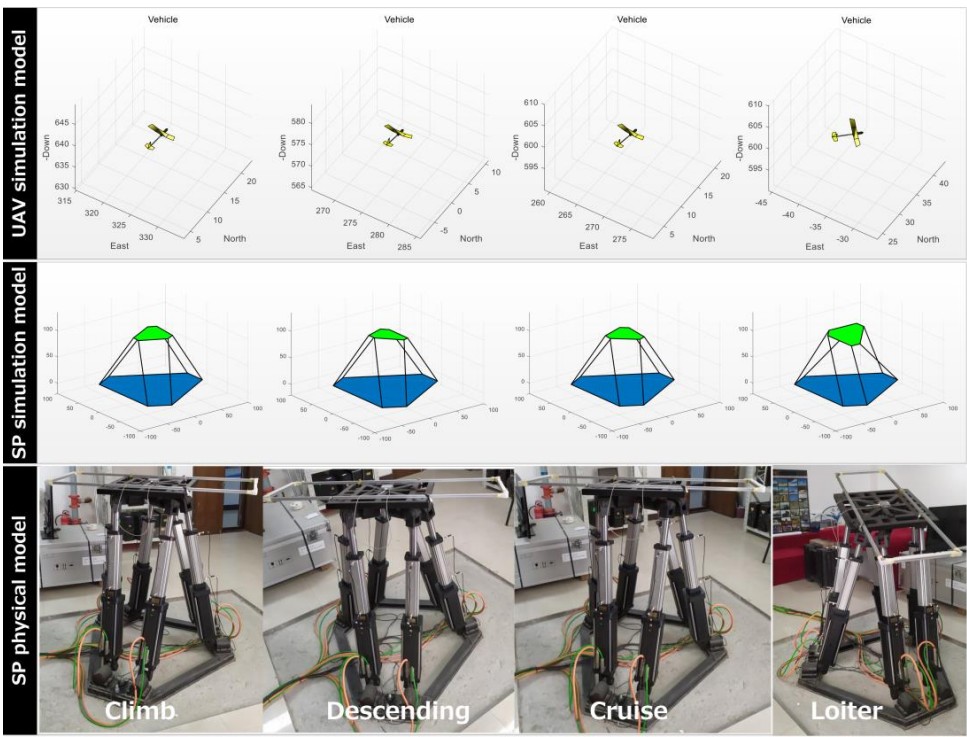

**Figure 19.** UAV-SP co-simulation in different flight phases.

The combination of three modes can verify the flight envelope and boundary state, calibrate the measurement error of the sensor and the result of state estimation, and test the feasibility of the whole system; the angular velocity mode can also be regarded as a combination of multiple attitude modes. The angular velocity and attitude modes can be used to simulate the states in the envelope, and the acceleration mode is used to simulate the boundary and oscillation divergence states. Table 9 is a summary of the characteristics of three modes.

**Table 9.** Summary of the characteristics of three different experiment modes.

| Experiment Mode | Scheme | Experiment Process | Purpose | Calibration | Characteristic |
|---|---|---|---|---|---|
| Acceleration | Open loop | Flight envelop boundary | Structural overload test | Accelerometer/Gyro | Dynamic process |
| Angular velocity | Open and closed loop | Typical flight modes | Flight mode verification | Gyro/Attitude angle | Dynamic process |
| Attitude | Open and closed loop | All mission status points | Control logic verification | Gain parameters | Static point |

### 4. Field Flight Test

The above simulation process is a preliminary and a laboratory test of the whole system; due to the lack of position and velocity information, a field test is necessary for the whole system verification. The target mission areas of the UAV are Qiangtang and Hoh Xil, Tibet, the flight conditions are high altitude, low temperature, low pressure, and limited flight area, and the field test flight is divided into two stages of low and high altitude to fully verify the design scheme and compare the difference caused by high altitude in detail. To ensure the experiment consistency, the power and payload conditions of UAV are the same in the two experiments. The low-altitude test site was located in Xi'an (34.033 E, 109.100 N) with a height of 635 m, the time was November 29, 2018, 13:00, and the temperature was 6 °C; the high-altitude test was in Qiangtang (31.988 E, 87.317 N) with a height of 4554 m, and it was at the time the same as the numerical simulation. The temperature difference due to the altitude difference was approximately −15 °C during winter, the take-off speed increased from 8.5 to 11.5 m/s, and the cruise speed increased from 10 to 11.5 m/s.

*4.1. Flight Results*

The field tests at low and high altitudes are shown in Figure 20. Figure 21 shows the flight data in the cruise phase of a four-sided rectangular mission route. The UAV has no landing gear and fuselage touches the ground directly, and it takes off from the car by hand, and the state bus for communication with the ground station includes speed, height, position, attitude, and commands. According to Figure 21a,b, the accuracy of the longitudinal parameters of high and low altitude is closed, the height error precision is nearly 8 m, the airspeed is nearly 4 m/s, and the pitch angle is nearly 3 degrees. It is found that the wind speed at high altitude is close to 10 m/s, while that at low altitude is 3 m/s, which causes an obvious deviation in heading accuracy. The error of yaw angle in high altitude is 10.3 degrees, which is twice of that in the low area. In the state estimation module, the first two stages of the estimation aimed at improving the accuracy of the attitude for the UAV without aileron, a stable roll angle estimation, and control ensuring that the system has the ability of autonomous flight. The roll angle error accuracy of the two altitudes is approximately 6.5 degrees, and the trajectory accuracies are respectively 23 m and 44 m, which meet the design requirements.

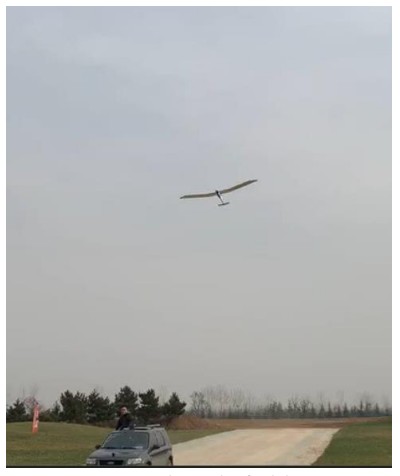
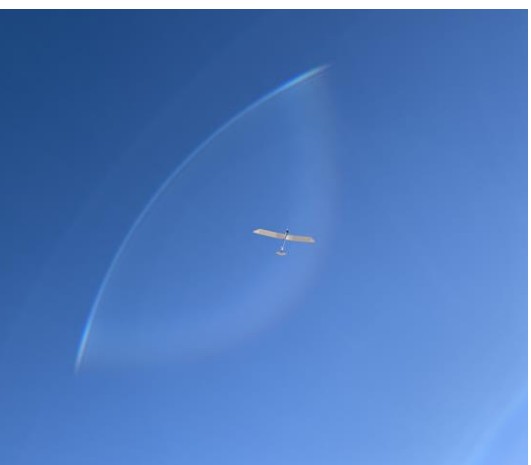

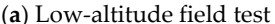

(**a**) Low-altitude field test          (**b**) High-altitude field test

**Figure 20.** Real flight test in low and high altitudes.

The flight parameters of four typical phases are shown in Table 10. The increase in altitude reduces the air density, the airspeed increases by 10%, the motor power increases by 40%, and the attitude amplitude is twice the low altitude. The reduction of propeller thrust and the increase of take-off speed will influence the take-off mode, so the car launching take-off is the only mode. Since the light intensity in the plateau region is about twice as strong as that in the low altitude, the endurance is approximately

the same. The results show that the UAV has the capability of autonomous flight at high attitude, the altitude precision is less than 10 m, and the positioning precision is less than 50 m/km$^2$.

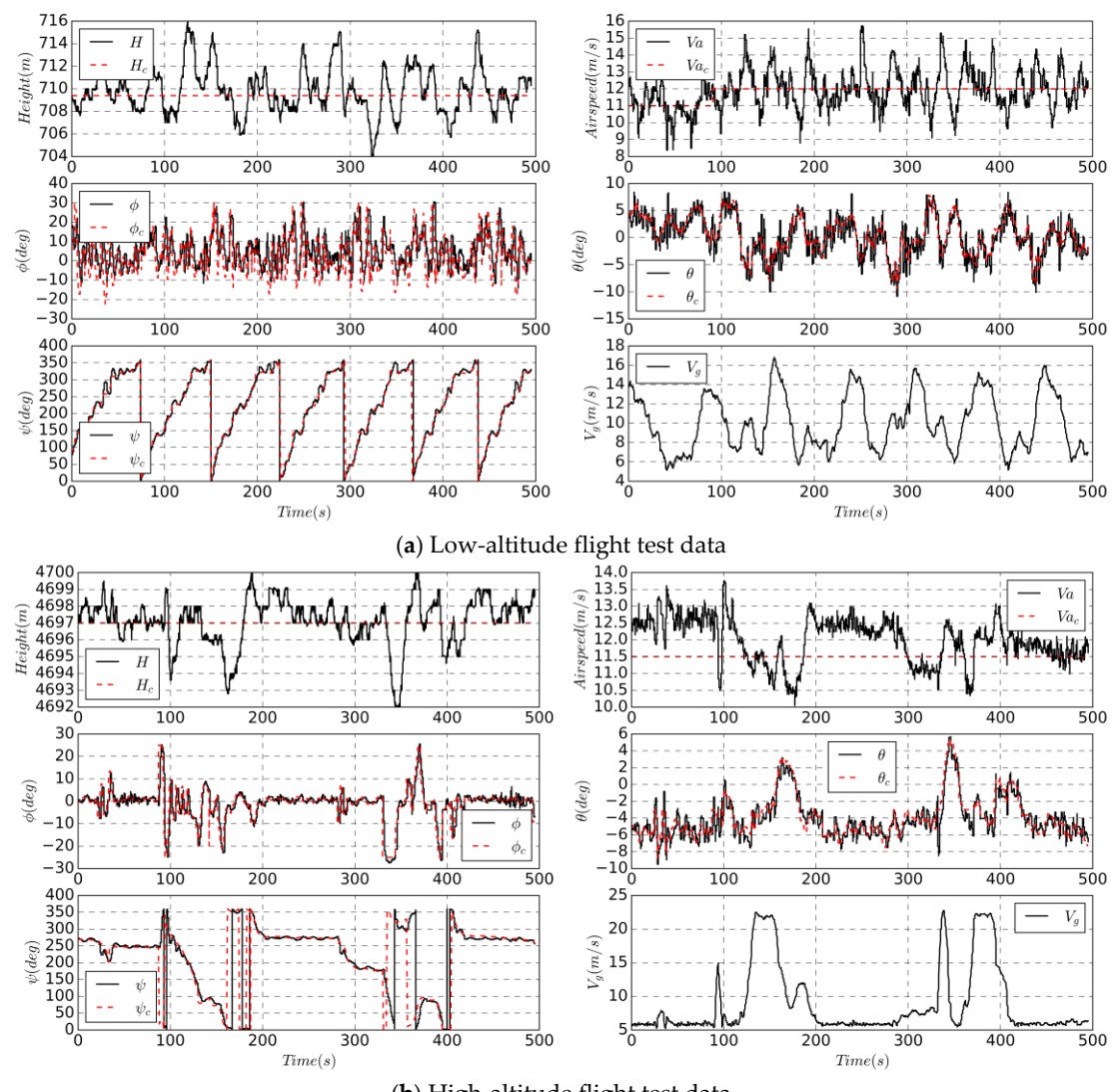

(**a**) Low-altitude flight test data

(**b**) High-altitude flight test data

**Figure 21.** Flight data in the cruise phase of low and high altitude.

**Table 10.** Flight parameters for four typical phases at different altitudes.

| Phases | Height (m) | Airspeed (m/s) | Pitch Angle (deg) | Roll Angle (deg) | Motor Power (W) |
|--------|-----------|----------------|-------------------|------------------|-----------------|
| Climb | [638 741] | 12.55 | 3.82 | 2.29 | 142.7 |
| | [4600 4700] | 11.27 | 4.67 | 1.09 | 200.3 |
| Cruise | 710.4 | 10.62 | −1.58 | −0.56 | 47.06 |
| | 4697 | 11.76 | −4.19 | −1.08 | 78.4 |
| Loiter | 781.9 | 10.9 | −1.34 | −8.2 | 66.4 |
| | 4629 | 11.0 | −3.26 | −3.41 | 80.2 |
| Descend | [780 653] | 11.7 | −4.6 | 2.9 | 31.1 |
| | [4787 4694] | 13.47 | −9.87 | 0.64 | 32.5 |

*4.2. Model Calibration*

Simulation and model correction are mutually reinforcing processes. The former is to predict the final result through a general rule, and the latter is a reverse process to ensure that the prediction result

is more in line with the actual situation [29]. After the field test, the flight data can be used to verify the correctness of the system model and to calibrate the key parameters of the modeling. Using the parameters in the cruise state as a reference, the flight data can be used to calibrate the longitudinal lift-drag model, sensor, and state estimation module.

The longitudinal lift, drag coefficient curves, and elevator influence are shown in Figure 22, and the effect of the elevator on lift and drag is calculated by its derivatives, $C_L{}^{\delta e}$ and $C_D{}^{\delta e}$. The design point of the maximum lift-to-drag ratio is 0 to 4 degrees angle of attack, and the UAV cruises in this state. Considering the cost of the experiment, no airflow angle sensor is installed in the control system.

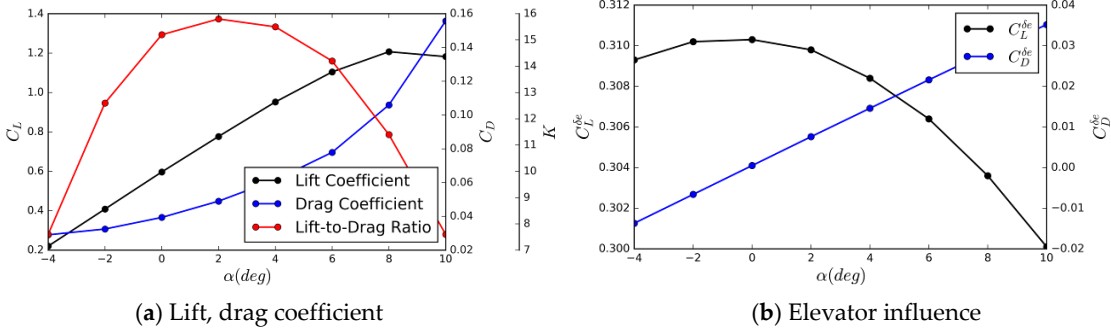

(**a**) Lift, drag coefficient          (**b**) Elevator influence

**Figure 22.** Longitudinal aerodynamic coefficient.

Based on the windless assumption, when the UAV is in the vertical plane, the relationship among climb angle $\gamma$, pitch angle $\theta$, and angle of attack $\alpha$ is $\gamma = \theta - \alpha$, and in the cruise phase, $\gamma \approx 0$, leading to $\theta \approx \alpha$, and the longitudinal balance equation is given

$$\begin{cases} T_P \cos \alpha = D \\ T_P \sin \alpha + L = mg \\ M + T_P l_z = 0 \end{cases} \tag{20}$$

where $L$, $D$, and $M$ are the lift, drag, and pitching moment, and $l_z$ is the vertical distance from the center of the propeller to the center of gravity. Aerodynamic force can be calculated by the force coefficient, dynamic pressure, and wing area, and the force coefficient is a function of the UAV flight state, $C_i = f_i(H, V_a, \theta, \delta_e)$. The cruise state parameters of the two flights and the longitudinal aerodynamic force coefficients are shown in Table 11. The force coefficient is divided into *model* and *real*; the former is obtained from the Computational Fluid Dynamics (CFD) results, and the latter is the reverse calculation result based on the balanced state. Then, the *real* data can be used to calibrate that of the *model*. The results show that the *model* lift coefficient is 14.5% larger and the drag coefficient is 20% smaller than that of the *real*. With the increase of height, the lift–drag ratio decreases by 27%.

**Table 11.** The summary of UAV cruising parameters at low and high altitude.

| $H$ m | $V_a$ m/s | $\theta$ deg | $\delta_e$ deg | $\delta_t$ | $T_P$ N | $C_L$ Model | $C_D$ Model | $C_m$ Model | $C_L$ Real | $C_D$ Real | $C_m$ Real | $L/D$ |
|---|---|---|---|---|---|---|---|---|---|---|---|---|
| 710 | 10.6 | −0.28 | 6.9 | 0.37 | 2.43 | 0.527 | 0.036 | 0.008 | 0.405 | 0.041 | 0.034 | 9.9 |
| 4697 | 11.7 | 1.15 | 3.74 | 0.52 | 4.46 | 0.71 | 0.045 | −0.02 | 0.67 | 0.08 | −0.07 | 7.8 |

Except for the basic aerodynamic force correction, the dynamic error of the IMU can also be corrected by the real measurement during the flight. A Finite Impulse Response (FIR) filter is applied to filter the measurement data as a reference state, and the noise and bias in the model can be calibrated

by the relationship between the range and mean value of the measured and filtered data, shown as follows

$$\eta_{correct} = \frac{X_m|_{min}^{max} - X_f|_{min}^{max}}{2}$$
$$\beta_{correct} = E(X_m) - E(X_f)$$

(21)

where the subscripts $m$ and $f$ represent the measured and filtered data. The measurement noise and bias of the IMU are shown in Table 12, and the error and noise in Table 2 are the same in the *xyz* direction, but the acceleration in the *x* direction and the angular velocity in the *y* direction are obviously greater than that of the other two directions, and the noise in the corresponding direction is also greater. With the increase of altitude, the noise increases and the deviation decreases, which can be regarded as a function of altitude and airspeed, but the sensor bias is almost constant and it can be regarded as a function of temperature.

**Table 12.** The statistics of IMU measurement noise and bias.

| | | Accelerometer (m/s$^2$) | | | Gyroscope (deg/s) | | |
|---|---|---|---|---|---|---|---|
| | | **x Direction** | **y Direction** | **z Direction** | **p** | **q** | **r** |
| **Low altitude** | Noise | 0.0818 | 0.0193 | 0.2514 | 4.1424 | 5.915 | 5.146 |
| | Bias | 0.00424 | 0.00315 | −0.0875 | −0.1074 | 0.0383 | 0.3928 |
| **High altitude** | Noise | 0.1286 | 0.029 | 0.1211 | 5.427 | 7.824 | 4.522 |
| | Bias | 0.0003 | 0.00143 | −0.0372 | −0.0034 | 0.0031 | −0.0417 |

## 5. Conclusions

The paper presents an evolutionary stage in the development of an aileron-less low-cost LALE solar-powered UAV from the concept of flight control design to the real-life field test. The modeling process is complete, and the component-level modeling method of energy and flight control system can take more parameters and influences into account, which has reference significance for small solar aircraft.

A complete verification process from numerical simulation to HILS to field test shows that the accuracy sequence of state parameters and energy balance process of the simulation process are consistent with the actual flight. Based on the simulation model and Stewart platform, a novel SPHIL experiment is established, and three experiment modes of acceleration, angular velocity, and attitude are designed to test the inner and boundary states of the flight envelope. Besides, the HILS system can be used to calibrate the measurement of the low-cost sensor, verify the accuracy of the control command, and reproduce the typical flight states.

The flight test at different altitudes is the verification of modeling and simulation, which shows that the UAV scheme meets the design requirements. Combined with the filtered results, the aerodynamic force coefficients and sensor measurement error can be further calibrated. As the altitude increases, the control accuracy of the attitude angle is twice that of the low altitude, the precision of the heading angle is unchanged, the trajectory accuracy is increased by a factor of approximately two, the cruising power increases by 66%, the model lift coefficient is 14.5% larger, the drag coefficient is 20% smaller, and the measurement noise of the sensor increases. Future work will focus on the exploration of long-endurance hardware reliability and the environmental adaptability of the UAV.

**Author Contributions:** A.G. and Z.Z. conducted the whole system modeling; X.Z. (Xiaoping Zhu) contributed the control system modeling; Z.Z. and X.Z. (Xiaoping Zhu) performed the experiments; Z.Z. and X.Z. (Xiaoping Zhu) analyzed the experiment results; Y.D. conducted the Hardware-In-the-Loop simulation experiment; A.G. and X.Z. (Xin Zhao) wrote the paper. All authors have read and agreed to the published version of the manuscript.

**Funding:** This research was funded by the Equipment Pre-research Project (No. 41411020401), the National Key R&D Program in Shaanxi Province (2018ZDCXL-GY-03004), and the Innovation Program of Research Institutions (No. TC2018DYDS24).

**Conflicts of Interest:** The authors declare no conflict of interest.

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
