# Peer review of "Automatic Control and Model Verification for a Small Aileron-Less Hand-Launched Solar-Powered Unmanned Aerial Vehicle"

_electronics, doi:10.3390/electronics9020364_

Round 1

Reviewer 1 Report

It is a detailed research paper, I would recommend a series of tests in the wind tunnel to confirm the maneuvering performances.

Reviewer 2 Report

In the paper, the authors presented a detailed AUV's modelling as well as the experiments based on Stewart-platform and flight tests at different altitudes. The obtained results were scrupulously presented and thoughtfully analyzed. The only few remarks to the submitted manuscript are presented below.
Line 71- The sentence "The specific contributions... " does not suit the rest of the text.
Line 81-86 - The sentences should be corrected.
Line 95 - Table 1 and Table 2 present different values of Battery energy density.
Line 138 - Why is the engine system model formulated "According to the above experiment results". Additionally, D and V were not introduced in the text and J instead of Jp was used in the equations.
Line 204 - In Figure 6, the values were not meausered but simulated. The legends should be improved.
Line 333 - The abbreviation HILS should be introduced at the first use.

Reviewer 3 Report

The Reviewer appreciated the article and only requires some minor modifications:

Pag.2 line 71. After the sentence “The specific contributions of this paper are:” a list with bullets is expected. Please, consider writing it in this way.

Pag.3 line 79. Please, consider to include a table with all the top level aircraft requirements and, separately, include a rationale for all of them. For instance, it is stated that the minimum temperature is -10°C and the take-off and landing altitudes is above 5000 m. However, using Earth atmosphere model, at 5000m the mean temperature is -17°C.

Pag.3 line 84. In the sentence “the nose in order to get a wild filed view as well as avoid touchdown collision” the word “wild” should be probably replaced with “wide”

Fig.2 could be improved. It is not easy to understand where the engine is. E.g. the propeller could be added.

Pag.5 Eq.3 Some variables are not introduced. E.g. speed, air density etc.

Pag.6 line 147 solar flux instead “solar flex”

Pag.7 line 175 “sensor combination” instead “senor combination”

Pag.9 line 212 senor instead of sensor

Pag.11 line 272 check reference to Fig.12 (maybe it is Fig.9b)

Pag.18 line 426 check reference to Table 4 (maybe it is Table 9)

Other misspelled words and some sentences should be corrected. The reviewer recommend to proofread the text.

Reviewer 4 Report

I think that this paper describes the entire process of controlling an UAV with electric propulsion. I highlight the models used for the energy system, which have been adjusted based on experimental data. Furthermore, the idea of using a Stewart-Platform as a first step before carrying out flight tests is very appropriate and interesting. Despite that, I have some comments on the manuscript:

I think that the use of first person in the Introduction and Abstract is not appropriate. The authors decided to control an aileron-less aircraft. The manuscript should include an analysis of the state of the art on this kind of aircraft, which current aircraft use this technology and even which control strategies are chosen for them. In my opinion, this information should be included in the Introduction. I do not understand why the control requirements are those described in the first paragraph of section 2. Perhaps, a description of the mission for which the UAV was designed or selected could help readers to better understand this decision. There is a typo in line 272, the corresponding Figure is not number 12. There is no information on how the aircraft model has been determined in order to perform the simulations and the theoretical design of the autopilot. Is the model based on stability derivatives? How have they been estimated? How have the inertia properties been determined?
